# Identification of *Aspergillus niger* Aquaporins Involved in Hydrogen Peroxide Signaling

**DOI:** 10.3390/jof9040499

**Published:** 2023-04-21

**Authors:** Thanaporn Laothanachareon, Enrique Asin-Garcia, Rita J. M. Volkers, Juan Antonio Tamayo-Ramos, Vitor A. P. Martins dos Santos, Peter J. Schaap

**Affiliations:** 1Laboratory of Systems and Synthetic Biology, Wageningen University & Research, 6708 WE Wageningen, The Netherlands; enrique.asingarcia@wur.nl (E.A.-G.); j.m.volkers@nvwa.nl (R.J.M.V.); 2Enzyme Technology Laboratory, Biorefinery and Bioproduct Technology Research Group, National Center for Genetic Engineering and Biotechnology, 113 Thailand Science Park, Khlong Luang, Pathumthani 12120, Thailand; 3Biomanufacturing and Digital Twins, Wageningen University & Research, 6708 PB Wageningen, The Netherlands; vitor.martinsdossantos@wur.nl; 4ITENE Research Center, Industrial Biotechnology Area, C/Albert Einstein 1, 46980 Paterna, Valencia, Spain; ja.tamayoramos@gmail.com; 5UNLOCK Large Scale Infrastructure for Microbial Communities, Wageningen University & Research, Delft University of Technology, 6708 WE Wageningen, The Netherlands

**Keywords:** *Aspergillus niger*, aquaporin; X-intrinsic protein, XIPs, yeast expression, hydrogen peroxide

## Abstract

*Aspergillus niger* is a robust microbial cell factory for organic acid production. However, the regulation of many industrially important pathways is still poorly understood. The regulation of the glucose oxidase (Gox) expression system, involved in the biosynthesis of gluconic acid, has recently been uncovered. The results of that study show hydrogen peroxide, a by-product of the extracellular conversion of glucose to gluconate, has a pivotal role as a signaling molecule in the induction of this system. In this study, the facilitated diffusion of hydrogen peroxide via aquaporin water channels (AQPs) was studied. AQPs are transmembrane proteins of the major intrinsic proteins (MIPs) superfamily. In addition to water and glycerol, they may also transport small solutes such as hydrogen peroxide. The genome sequence of *A. niger* N402 was screened for putative AQPs. Seven AQPs were found and could be classified into three main groups. One protein (AQPA) belonged to orthodox AQP, three (AQPB, AQPD, and AQPE) were grouped in aquaglyceroporins (AQGP), two (AQPC and AQPF) were in X-intrinsic proteins (XIPs), and the other (AQPG) could not be classified. Their ability to facilitate diffusion of hydrogen peroxide was identified using yeast phenotypic growth assays and by studying AQP gene knock-outs in *A. niger.* The X-intrinsic protein AQPF appears to play roles in facilitating hydrogen peroxide transport across the cellular membrane in both *Saccharomyces cerevisiae* and *A. niger* experiments.

## 1. Introduction

*Aspergillus niger* is a potent cell factory that is successfully employed to produce organic acids, including citric-, oxalic-, and gluconic acids [1,2]. Citric- and oxalic acids are intracellularly produced and actively secreted [3,4,5,6]. In contrast, gluconic acid is the product of an extracellular conversion of glucose in a collaboration of three extracellular enzymes: glucose oxidase (GOx), lactonase, and catalase [7]. At an extracellular pH between 4.5 and 6.5 and in the presence of sufficient amounts of oxygen, glucose oxidase catalyzes the oxidation of glucose to gluconolactone and hydrogen peroxide. The co-expressed lactonase subsequently converts gluconolactone to gluconate [7,8].

Recently, the role of hydrogen peroxide by-products in the regulation of the *A. niger* (GOx) expression system has been elucidated [8]. Hydrogen peroxide is a cell-damaging reactive oxygen species (ROS), and the role of the co-induced extracellular catalase is to decompose the toxic levels of the hydrogen peroxide by-product into water and oxygen. Hydrogen peroxide is, however, also a signaling molecule controlling metabolic pathways [9,10]. In fungi, its functions as an essential second messenger involved in the regulation of the redox level and downstream signaling pathway [11]. In *A. niger*, the hydrogen peroxide by-product acts as a second messenger triggering, in a feed-forward loop, the expression of the GOx system [8], which suggests facilitated diffusion of hydrogen peroxide across the plasma membrane mediated by aquaporins.

Aquaporin proteins (AQP) are transmembrane protein channels belonging to the major intrinsic proteins (MIPs) superfamily and have been detected in genomes of bacteria, fungi, plants, animals, and humans [12]. Structurally, these proteins commonly comprise six transmembrane domains connected by five loops, with the N- and C-terminal ends located in the cytoplasm and highly conserved NPA (asparagine-proline-alanine) motifs at the center of the structure [13]. They act as channels to facilitate the selective transport of water, neutral solutes, and small uncharged molecules across cellular membranes [14,15,16]. Some AQPs, such as AQP3 [17] and AQP8 [18] of human cells, AQP8 of *Botrytis cinerea* [19] and, TIP1;1 and TIP1;2 of *Arabidopsis thaliana* [18], have been shown to mediate hydrogen peroxide uptake. 

Currently, fungal MIPs have been classified into three groups: orthodox AQPs, aquaglyceroporins (AQGP), and X-intrinsic proteins (XIPs) [20,21]. Particularly, the structure, function, and regulation of the fungal AQPs have been less studied. Very few fungal AQPs have been functionally characterized, such as *Saccharomyces cerevisiae* [22], *Laccaria bicolor* [23], *Glomus intraradices* [24], and *Aspergillus glaucus* [25]. Additionally, from fungal genome databases, many more putative fungal AQPs can be identified [22,26]. 

By screening the genome sequence of *A. niger* strain ATCC 64974 (N402), seven putative AQPs were identified. Experiments were designed to observe aquaporin-facilitated hydrogen peroxide transport. Yeast strains expressing individual AQPs were evaluated by monitoring growth under various hydrogen peroxide concentrations and by fluorescence assays. Apart from the yeast phenotypic growth assays, *A. niger* AQP knock-out strains were constructed, and AQP transcript levels were monitored in wild-type and AQP knock-out strains before and after the addition of hydrogen peroxide to the medium. 

From the yeast experiments, three AQPs were identified with a potential role in hydrogen peroxide membrane transport. Integration of these results with *A. niger* expression data suggests that AQPF may play an important role in facilitating hydrogen peroxide transport across the *A. niger* cellular membrane.

## 2. Materials and Methods

### 2.1. Strains and Media

All strains used in this study are listed in Table 1. *A. niger* ATCC 64974 (N402;*cspA1*) [27] was used as a control strain. *A. niger* strain MA164.9 (*kusA*::DR-*amdS-*DR, *pyrG*^-^) is a descendant of N402 [28] and was used to construct AQP knock-outs using the *pyrG* gene from *A. oryzae* as a selectable marker gene. The pWay-*pyrA* plasmid was used as a control for transformation. *A. niger* strains were maintained on a complete medium (CM) containing 1 g/L casamino acid, 5 g/L yeast extract, 1% glucose, 20 mL/L ASPA + N, 1 mL/L, Vishniac solution, and 1 mM MgSO_4_ at 30 °C [29].

The *S. cerevisiae* strain BY4741 (*MATa his3*Δ*1 leu2*Δ*0 met15*Δ*0 ura3*Δ*0*) was used in this study for the functional validation of the aquaporin genes [30]. The strain was grown at 30 °C in a YPD medium composed of 1% (*w*/*v*) yeast extract, 2% (*w*/*v*) bactopeptone, and 2% (*w*/*v*) glucose. To prepare electro-competent cells, the strain BY4741 was prepared as described by Suga and Hatakeyama [31]. In the growth experiments, the *S. cerevisiae* strains carrying *A. niger* aquaporin genes were maintained on a solid Yeast Nitrogen Base (YNB) minimal medium containing 0.7% (*w*/*v*) BD Difco YNB without amino acids, 1% (*w*/*v*) glucose, 0.003% (*w*/*v*) L-leucine, 0.002% (*w*/*v*) L-methionine, 0.002% (*w*/*v*) L-histidine, and 1.5% (*w*/*v*) agar.

### 2.2. Identification of A. niger Aquaporin Sequences

The *A. niger* N402 genome (Aniger_ATCC64974_N402) with accession number GCA_900248155.1 was used to search for aquaporins. The presence of the MIP domain in the resulting putative sequences was identified by Pfam [32], PROSITE [33], and SMART [34]. Transmembrane domains were determined by the HMMTOP transmembrane topology prediction server [35]. The N402 aquaporins were aligned with other fungal aquaporins, selected from Verma et al. [21] using ClustalX [36]. The alignment was applied to the phylogenetic analysis conducted using MEGA version 7.0 [37]. The phylogenetic tree was prepared by the neighbor-joining method with 1000 bootstrap replications referring to the reliability tests of an inferred tree [38].

### 2.3. Heterologous Expression of A. niger Aquaporins in S. cerevisiae

Spores of *A. niger* N402 were grown in CM and incubated at 30 °C for 40 h. After that, the mycelium was harvested and then used in the RNA extraction, following the protocol provided in the Maxwell^®^ 16 LEV simplyRNA Cells Kit (Promega, Madison, WI, USA). RNA concentrations were measured by NanoDrop, and purified RNA was stored at −80 °C. The purified RNA was used to synthesize cDNA following the RevertAid H Minus First Strand cDNA Synthesis Kit (ThermoFisher, Waltham, MA, USA). The reaction mixture was incubated at 42 °C for 1 h and then terminated by heating at 70 °C for 5 min. The cDNA was stored at −20 °C.

The aquaporin genes were cloned into an overexpression vector using the Yeast ToolKit (YTK) described by Lee et al. [39]. In short, the *A. niger* aquaporin gene sequences were first inspected to find internal restriction sites (IRS) for the enzymes used in the YTK. Specific primers (Appendix A) were designed to remove these IRS by introducing synonymous substitutions. The cDNAs of *A. niger* N402 and *S. cerevisiae* BY4741 were used as templates for the amplification of the aquaporin genes and the *FPS1* gene, respectively. Q5 high-fidelity DNA polymerase (New England BioLabs, Ipswich, MA, USA) was used in the PCR reaction, following the manufacturer’s protocol. Amplified fragments were purified by NucleoSpin Gel and PCR Clean-up kit (Macherey-Nagel, Düren, Germany), and DNA concentrations were measured by NanoDrop. 

The aquaporin genes were cloned into the YTK entry vector, according to Lee et al. [39]. Each restriction-ligation product was transformed into *E. coli* DH5α and grown at 37 °C on LB agar supplemented with 25 µg/mL of chloramphenicol.

Plasmids were isolated from an overnight culture of single colonies using GeneJET Plasmid Miniprep Kit (Thermo Scientific, Waltham, MA, USA) according to the manufacturer’s protocol. The integrity of the plasmids was checked by restriction digestion analysis and confirmed by Sanger sequencing (GATC Biotech, Landkreis Ebersberg, Germany). 

To assemble the yeast expression plasmids, the restriction-ligation reactions contained 20 fmol of each DNA module (promoter pRPL18B, aquaporin gene, and terminator tPGK1) and 40 fmol of backbone plasmid pMV009. After the reaction, the gene-cassettes were transformed into *E. coli* DH5α and grown at 37 °C on LB agar supplemented with 100 µg/mL of ampicillin. A plasmid map is shown in Appendix A. The yeast expression plasmids were transformed into *S. cerevisiae* BY4741 electrocompetent cells as described by Suga et al. [31,40] (Table 1).

### 2.4. Growth and Hydrogen Peroxide Sensitivity Assays in S. cerevisiae

Yeast cells were grown in the selective YNB medium at 30 °C and 250 rpm for 18 h. Overnight cultures were diluted in fresh medium to an OD_600_ of 0.1, and subsequently, serial dilutions were made. Five-µL of the diluted cell suspensions was spotted onto the solid selective YNB medium containing different concentrations of hydrogen peroxide (0–2 mM) and incubated at 30 °C. Growth and survival were scored after 6 days of incubation. For Ag^+^ treatments, the cell suspensions were spotted on the solid selective YNB medium supplemented with either no or 1 mM hydrogen peroxide and various concentrations of AgNO_3_ (0–15 µM). The plates were incubated at 30 °C and then incubated for 6 days before growth and survival were scored. The experiments were performed in triplicate.

The transport of hydrogen peroxide was studied using a fluorescence-based assay adapted from Bienert et al. [18]. Yeast cells were pre-cultured on the solid selective YNB medium for 2 days. Three-mL liquid cultures were inoculated with a single colony and supplemented with 2′,7′-dichlorodihydrofluorescein diacetate (DCFHDA; Sigma-Aldrich, St. Louis, MO, USA) with a final concentration of 1 µM. Cells were grown at 30 °C and 250 rpm in darkness overnight to an OD_600_ of 1.6. The cells were washed five times in 20 mM HEPES buffer, pH 7.0, and finally resuspended in HEPES buffer at an OD_600_ of 1.4. The yeast cell suspensions were aliquoted into 96-well plates with 200 µL per well. The suspensions were followed over time at room temperature using a microplate reader with fluorescent mode (Synergy^TM^ Mx Monochromator-Based Multi-Mode Microplate Reader; BioTek, Broadview, IL, USA) at excitation/emission of 492/527 nm. After the initial (t = 0) measurements, hydrogen peroxide was automatically dispensed in the wells at concentrations of 0.1, 0.5, and 1.0 mM. The OD_600_ and fluorescence intensity were recorded every minute within 2 h. A heatmap was generated from an average fluorescent intensity per OD_600_ from the experiment with two biological and two technical replicates.

### 2.5. Construction of A. niger AQP Knock-Out Strains

The AQP knock-out strains were constructed using the split-marker approach [29,41]. Three selected aquaporin genes: *aqpD*, *aqpE*, and *aqpF*, were individually deleted from the genome of the *A. niger* MA169.4 strain, which is defective in the Non-Homologous End-Joining (NHEJ) pathway through a transiently silenced *kusA* gene [41]. The preparation of DNA fragments and protoplasts and transformation steps were done according to Laothanachareon et al. [8]. The list of the primers used to construct the knock-out strains and to confirm the correct deletion of the genes by PCR is shown in Appendix A.

### 2.6. Transcriptional Analysis of Aquaporins in A. niger

Shake-flasks were used to monitor the expression of *A. niger* aquaporin genes upon hydrogen peroxide treatment. Using an inoculation of 1 × 10^6^ spores/mL, *A. niger* wild-type N402 and knock-out strains were pre-cultured for 18 h in minimal medium (MM) containing 4.5 g/L NaNO_3_, 1.13 g/L KH_2_PO_4_, 0.38 g/L KCl, 0.38 g/L MgSO_4_.7H_2_O, 2 g/L casamino acid, 1 g/L yeast extract, 1 mL/L Vishniac trace element solution, and supplemented with 50 mM fructose as a carbon source, at 30 °C and 200 rpm. Mycelium was harvested, rinsed with water, and then transferred to fresh MM supplemented with 50 mM fructose at an initial pH of 6.0. The experiments were performed in the presence of various concentrations of hydrogen peroxide (between 0 and 75 mM, see Section 3) and with various incubation times (between 0 and 3 h, see Section 3). After incubation, the mycelium samples were taken for RNA isolation, quickly washed, and then dried with a single-use towel, snap-frozen with liquid nitrogen, and stored at -80 °C until further processing. In all cases, two biological replicates with three technical replicates per condition were studied.

RNA was isolated from mycelium, as described by Sloothaak et al. [42]. Quantitative realtime PCR (RT-qPCR) and calculations were executed following the protocols and instrument setup as previously described by Mach-Aigner et al. [43]. Primer sequences are listed in Appendix A. Cycling conditions and control reactions were as described previously by Steiger et al. [44]. 

The histone H4-like transcript (ATCC64974_101030, ATCC 1015 gene ID 207921) was used to normalize the RT-qPCR expression data. The uninduced state (no addition of hydrogen peroxide) was used to compare expression levels.

## 3. Results

### 3.1. The Genome of A. niger Harbors at Least Seven Putative aqp Genes

Using sequence similarity methods and by comparing the conservation of encoded structural protein features, seven genes were identified in the *A. niger* genome sequence and named *aqpA*-*aqpG* (Table 2 and Appendix A). The length of the deduced protein sequences ranged from 250 to 617 amino acids. All deduced proteins have the MIP protein family signatures with six transmembrane domains except for AQPC and AQPG, presenting only five domains. The presence of the asparagine-proline-alanine sequences (NPA motifs), highly conserved in the aquaporin water channel family, was also analyzed (Appendix A). Two NPA motifs were found in AQPA and AQPD, one in AQPB and AQPF, whilst AQPC, AQPE, and AQPG do not contain this motif. Unusual NPA substitutions and alternative NPA-like loops were found in some of the sequences. The previously reported substitution NPS (asparagine-proline-serine) was found twice in AQPE and once in AQPG. The rest of the present NPA-like loops consist of NP- and N-A residues only (Appendix A). Exceptionally, AQPF showed only one single NPA motif. Taken together, protein sequence analyses suggested that the seven selected genes most likely encoded *A. niger* AQPs. 

### 3.2. Classification of A. niger AQPs

The seven AQPs of *A. niger* N402 were classified based on amino acid sequence similarity with other (fungal) AQPs (Figure 1). AQPD grouped with the γ-AQGPs, which are related to the β-AQGPs and the yeast FPS1-like AQGPs. AQPB and AQPE seem to be related to the Yfl054-like AQGPs. AQPA is the only representative of the orthodox AQPs and is in the same group as AQP8 of *Botrytis cinerea* (XP_001547129) [19]. The two remaining AQPs, AQPC and AQPF, appeared to belong to the X-Intrinsic Proteins (XIPs) clade. Ambiguously, AQPG appears to be related to the XIP class but is separate from the well-characterized groups.

### 3.3. AQP D, E, and F Facilitate Hydrogen Peroxide Import in Saccharomyces cerevisiae

*S. cerevisiae* transformants expressing *A. niger* AQPs were constructed to investigate their abilities in hydrogen peroxide import by indirect and direct experiments. In the indirect experiments, colony growth was monitored in the presence of 0–2 mM hydrogen peroxide (Figure 2). The *S. cerevisiae* strain used as a negative control contained the ‘empty’ pMV009 vector expressing a GFP control sequence, whereas the strain overexpressing the yeast aquaporin FPS1 worked as a positive control [13,19]. Due to their ability to facilitate the import of toxic amounts of hydrogen peroxide, the growth of the strains individually carrying the *A. niger aqpD*, *aqpE*, or *aqpF* genes was severely inhibited, starting at 1 mM hydrogen peroxide with a cell concentration below OD_600_ of 0.1 (Figure 2). However, the strains expressing *aqpA*, *aqpB*, *aqpC,* or *aqpG* showed a growth pattern similar to that displayed by the negative control. The positive control *FPS1* showed inhibition at 1.5 mM hydrogen peroxide. At the highest concentration, 2 mM, all strains, including the negative control, were affected, although some of them still appeared to show some growth when high cell concentrations were spotted.

A fluorescence-based assay [18] was used to directly follow the transport of hydrogen peroxide across the yeast plasma membrane over time. Here, 2′-7′- dichlorodihydrofluorescein diacetate (DCFHDA) was used as a detector. The DCFHDA fluorescent dye is cell-permeable ROS-sensitive. In its normal acetylated form, the dye can diffuse into the cells. Deacetylation traps the fluorochrome inside the cells and makes it susceptible to oxidation by ROS [18]. The fluorescence intensity of each AQP expressing yeast strain under the three different hydrogen peroxide concentrations of 0.1, 0.5, and 1.0 mM were compared over a 2 h period. The data were calculated as fluorescence intensity per OD_600nm_ using the condition without hydrogen peroxide to normalize the data (Figure 3). The strain transformed with the expression vector pMV009 was used as a negative control. At a concentration of 0.1 mM hydrogen peroxide, the expression of the *aqpD* gene-mediated transport of hydrogen peroxide into the cells already during the first hour, whereas higher hydrogen peroxide concentrations resulted in a slight decrease in the fluorescence signal. The *aqpA* and *aqpF*-expressing strains showed the same trend, but compared to the *aqpD*-expressing strain, their fluorescence signals were weaker. In the *aqpE* expressing strain, the fluorescence intensity barely increased during the first hour, although it started to be noticeable during the second hour. Obviously, the yeast strains carrying the *aqpB*, *aqpC*, or *aqpG* gene were not able to take up hydrogen peroxide.

Apart from hydrogen peroxide transport analysis under various hydrogen peroxide concentrations, we tested whether the specific aquaporin was responsible for the facilitated hydrogen peroxide transport. In this experiment, silver ions (Ag^+^) were used as an aquaporin inhibitor since it was reported that Ag^+^ could significantly inhibit the transmembrane flux of hydrogen peroxide by binding to cysteine or histidine residues in protein [18]. The *A. niger* AQPs contain 2–8 but structurally not conserved cysteine residues in their sequences that are targets for Ag^+^ inhibition (Appendix A). Increasing concentrations of Ag^+^ can cause a significant decrease in growth and cell viability [18]. In this study, the toxicity of Ag^+^ to yeast became evident at around 15 μM, which was higher than the previously reported 6 μM [45] and lethal at 30 μM. The hydrogen peroxide-induced growth phenotype of the yeast strains expressing *aqpD* and *aqpE* was considerably improved in the presence of Ag^+^ starting at 3.7 μM. In contrast, no restoration of growth was observed for the *aqpF* expressing strain. In the presence of Ag^+^, the negative control strain showed a similar growth phenotype in the absence and presence of hydrogen peroxide. This was also the case for AQP-expressing yeast strains previously found not to be growth inhibited by the addition of hydrogen peroxide (Figure 4). 

### 3.4. Transcriptional Analysis of AQPD, AQPE, and AQPF Single Knock-Out Strains

Single knock-out strains of *aqpD*, *aqpE*, and *aqpF*, actively facilitating hydrogen peroxide transport in yeast, were constructed. As hydrogen peroxide is both a reactive oxygen species known to cause damage to cellular components in high concentrations as well as a second messenger, we first analyzed AQP gene expression in the presence of various concentrations of hydrogen peroxide in the medium. Mycelium was incubated for 3 h with three different initial hydrogen peroxide concentrations. Fructose was used as a carbon source to avoid the generation of extracellular hydrogen peroxide by GOx. Upon the addition of hydrogen peroxide to the medium, the transcript levels of all AQP genes increased (Figure 5). As transcript levels of most AQP genes appeared to peak upon incubation with 10 mM hydrogen peroxide, a range of 0–10 mM hydrogen peroxide was chosen for transcriptional analysis of the AQP knock-out strains.

The growth phenotype of the three AQP knock-out strains was not noticeably different from the parental strain. The wild-type N402 and the three AQP knock-out strains were treated with a 10 mM hydrogen peroxide pulse and followed over time for 3 h by RT-qPCR. The *goxC* gene encoding glucose oxidase was used as a reporter gene to monitor hydrogen peroxide transport to the cell because, as a second messenger system, hydrogen peroxide can directly induce the expression of the glucose oxidase gene [8]. Due to the co-induction of extracellular CATR catalase activity [8], the hydrogen peroxide concentration will decrease over time. Relative transcript abundances (log10) of four genes: *aqpD*, *aqpE*, *aqpF*, and the *goxC* gene, were measured. The transcript levels of all genes increased during the first hour after the addition of hydrogen peroxide (Figure 6) and thereafter decreased. After 3 h, the system was returned to the uninduced state. Overall, we observed no or little cross-regulation. In the Δ*aqpD* background, transcript levels of the *aqpE* and *goxC* genes showed some increase after one hour, whereas the level of the *aqpF* gene was stable (Figure 6B). In the Δ*aqpE* strain, the expression pattern of the *goxC* gene was similar to that of the N402 control, while the two- and three-hour transcript levels of the *aqpF* gene were below the uninduced values (Figure 6C). The Δ*aqpF* strain behaved the same as N402, although the transcript levels of the *goxC* gene showed a lower expression level 1 h after the induction started (Figure 6D).

The responses of the wild-type and knock-out strains toward various initial concentrations of extracellular hydrogen peroxide were investigated further. Even though the expression levels of all genes were already decreasing 30 min after hydrogen peroxide addition, a fixed incubation time of 1 h was used for further studies because it showed significant differences in the expression of each gene. Samples were obtained from mycelium induced by initial hydrogen peroxide concentrations of 0 to 10 mM (Figure 7). In the N402 strain, all genes showed an increasing upregulation along with the increase in initial hydrogen peroxide concentration (Figure 7A). In the *aqpD* deletion strain, the lowest hydrogen peroxide concentration (0.2 mM) was not able to induce the expression of any of the studied genes; however, one hour after adding an initial hydrogen peroxide concentration of 10 mM increased transcription levels of *aqpE* and *goxC* were observed (Figure 7B). In the wild-type strain, *aqpD* gene transcript levels hardly respond to varying hydrogen peroxide concentrations (Figure 5 and Figure 7), and in yeast, AQPD mediates the transport of hydrogen peroxide into the cells already at a concentration as low as 0.1 mM hydrogen peroxide (Figure 4). Taken together, it appears that upon deletion of the *aqpD* gene, the GOx system is less sensitive to lower hydrogen peroxide concentrations. The transcription pattern of the *aqpD* and *aqpF* genes in the Δ*aqpE* background was not clearly dependent on the H_2_O_2_ concentration, while the expression pattern of the *goxC* gene was comparable to that observed in the N402 strain. The strain carrying a deletion in the *aqpF* gene was unable to induce the expression of all genes in the presence of the lowest hydrogen peroxide concentration. However, while the *aqpD* and *aqpE* genes recovered a normal expression pattern in the presence of 2 and 10 H_2_O_2_ mM, the *goxC* expression levels remained reduced compared to the wild-type strain (Figure 7D). 

## 4. Discussion

After the discovery of the first AQP over three decades ago, many members of the MIPs superfamily have been identified, cloned, and functionally studied [13]. However, information pertaining to the functional role of AQPs in ROS transport in fungal organisms is limited and currently absent for *A. niger*. In this study, we identified in the genome of *A. niger* ATCC64974 seven putative AQP genes. Comparative sequence analysis revealed that the encoded *A. niger* AQPs could be divided into three subclasses: one orthodox AQP (AQPA), three aquaglyceroporins (AQPB, AQPD, and AQPE), and two fungal XIPs (AQPC and AQPF). AQPG appeared to have only five transmembrane domains, and the conserved NPA-like sequences in loops B and E are absent in AQPG, and although AQPG showed sequence similarity with XIP AQPs, it could not be directly linked to a known AQP subclass (Figure 1).

The genomes of *Trichoderma* spp. also contain a similar number of aquaporin genes. The class distribution is, however, different. There are three orthodox AQPs, three AQGPs, and one XIP in *Trichoderma* ssp. [20]. In *A. niger*, only AQPA belongs to the orthodox AQP subclass. Orthodox AQPs are considered to be specific water channels [46] and therefore play important roles in cell osmoregulation [47,48]. Yeast cells expressing the *A. niger aqpA* gene can grow on plates containing high concentrations of hydrogen peroxide and/or in the presence of Ag^+^. However, *aqpA*-expressing yeast cells oxidized the ROS-sensitive fluorescent dye in the liquid medium, suggesting that AQPA encodes an orthodox water channel that can but does not efficiently facilitate the transport of hydrogen peroxide into the cells.

According to their amino acid sequence, AQPC, AQPD, and AQPE are aquaglyceroporins, which are expected to transport glycerol, urea, and other small solutes across cell membranes [14,49]. Based on sequence similarity, the *A. niger* aquaglyceroporins could be further subclassified: AQPB and AQPE are Yfl054-like aquaglyceroporins, while AQPD was classified as a γ-aquaglyceroporin. Yfl054-like aquaglyceroporins are found in both yeasts and filamentous fungi [22]. The Yfl054-like subgroup is characterized by an N-terminal extension of around 350 amino acids harboring the PVWSLXXPLPV motif and a C-terminal extension of around 50 amino acids. In filamentous fungi, this motif is partly conserved in *A. nidulans* and *Fusarium gramineum* Yfl054-like aquaglyceroporins [22]. PVWSLXXPLPV motif sequences were also found in the N-terminal extensions of the Yfl054-like aquaglyceroporins of *A. niger* (Appendix A). Although the functions of Yfl054-like aquaglyceroporins are still poorly described, they have been postulated as functional glycerol facilitators [50]. In addition, the Yfl054-like aquaglyceroporins play a specific role related to transmembrane solute fluxes, and they may be involved in regulatory processes through their long N-terminus based on their conservation of domain structure and sequence [22]. In our yeast experiments, only AQPE seemed to be able to facilitate hydrogen peroxide transport. 

Only one γ-AQGP, AQPD, was found in *A. niger*. This subgroup can be further subdivided into γ1 and γ2 AQGPs. The γ1 AQGP subclass is found in the species of *Mucoromycotina* [26], while the γ2 AQGP subclass is found in filamentous *Ascomycota* [21]. The AQPD of *A. niger* N402 is a member of the γ2-AQGPs. Conserved sequence motifs have been identified in loop B and E regions of γ2-AQGPs [21]. Accordingly, these motifs were also present in AQPD (Appendix A). In this study, AQPD was found to be able to transport hydrogen peroxide in yeast. Since the expression level of AQPD in N402 appeared to be identical under various hydrogen peroxide concentrations, it could act as to be constitutively expressed AQP facilitating hydrogen peroxide transport (Figure 7). 

AQPC and AQPF belong to the XIP subfamily. XIPs are commonly found in protozoa, plants, and fungi but not in bacteria and animals [51]. Fungal XIPs are frequently found in *Ascomycota*, *Basidiomycota*, and *Microsporidia* [25]. The XIPs have conserved motifs in loops B and E [21]. These sequence motifs were present in AQPF, whereas a single motif was found in loop B of AQPC (Appendix A). The biological functions and roles of fungal XIPs are still enigmatic. In plants, the XIPs are expected to facilitate the transfer of solutes such as urea, glycerol, hydrogen peroxide, boric acid, and ammonia because of their hydrophobic selectivity property that was characterized in *Populus* [52] and *Solanaceae* [53]. No transport of water could be observed by *Solanaceae* XIPs [53]. In contrast, two *Populus* XIPs apparently facilitate water transport [51]. The yeast phenotypic growth assays showed that only AQPF is able to facilitate hydrogen peroxide transport in yeast.

Hydrogen peroxide is a by-product of many intracellular and extracellular oxidative reactions, including the GOx system of *A. niger.* The *A. niger* GOx system uses the hydrogen peroxide by-product of the extracellular enzymatic conversion of glucose to gluconate as a second messenger to further induce the expression of the GOx system [8]. The facilitated diffusion of hydrogen peroxide into the cell could be explained by the expression of specific AQPs. Expression analysis of *A. niger* AQPs in the wild-type and AQP knock-out strains showed upregulation of all identified AQPs upon the addition of varying concentrations of hydrogen peroxide, while yeast phenotypic growth assays suggested that at least three *A. niger* AQPs: AQPD, AQPE, and AQPF, can transport hydrogen peroxide. Two of them, AQPD and AQPF, appear to play a more prominent role in the amplification of the hydrogen peroxide signal of the GOx system. A knock-out of AQPD showed a reduced sensitivity in the GOx system towards the lower hydrogen peroxide concentrations, and thus the constitutively expressed AQPD may play a role in the initial amplification of the GOx signal, while the AQPF knock-out has a major negative effect on *goxC* expression at the higher hydrogen peroxide concentrations.

## Figures and Tables

**Figure 1 jof-09-00499-f001:**
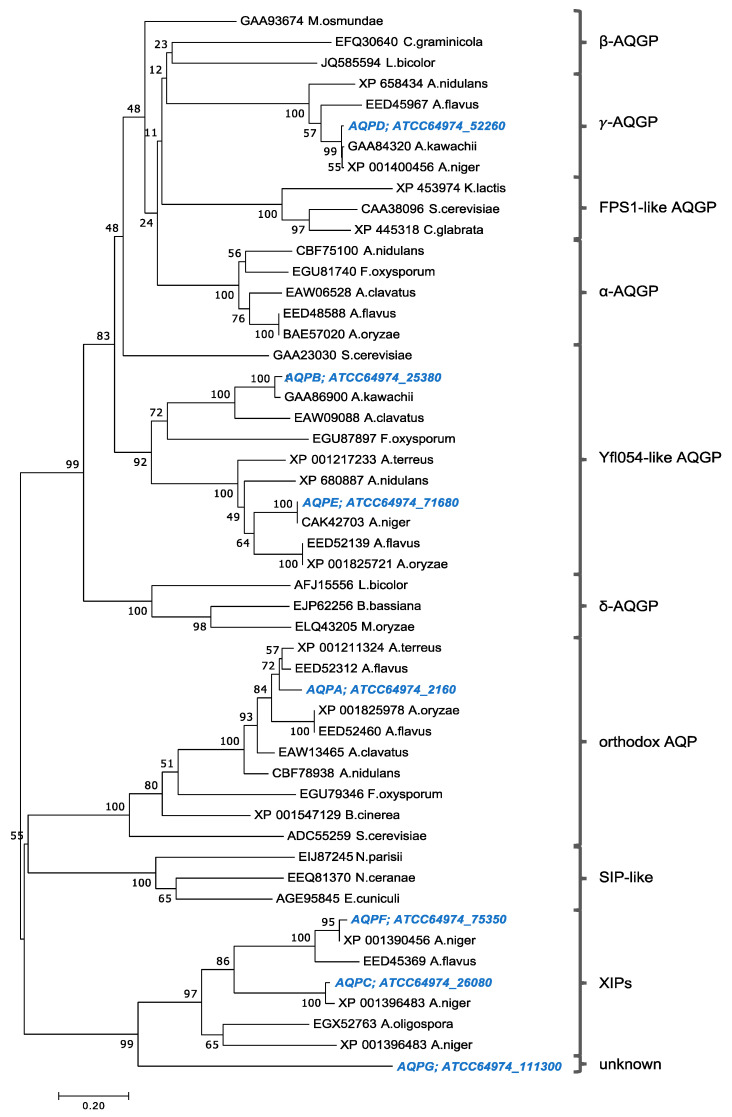
**Classification of *A. niger* N402 AQPs.** Sequences were aligned with ClustalX. The dendrogram was constructed by neighbor-joining analysis. *A niger* N402 sequences are indicated in blue. Number nodes indicate bootstrap support following 1000 iterations.

**Figure 2 jof-09-00499-f002:**
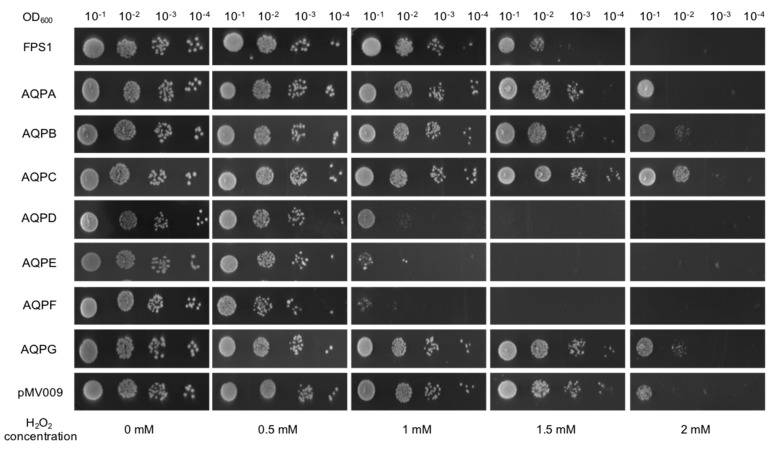
Growth and survival analysis of *S. cerevisiae* expressing *A. niger* AQPs in the presence of hydrogen peroxide. Overnight yeast cultures were diluted to an OD_600_ of 0.1, and subsequently, serial dilutions were done from this one. Five–µL of cell suspensions were spotted on agar plates containing various hydrogen peroxide concentrations (0–2 mM). Growth phenotype was recorded after incubating for 6 days at 30 °C. The data shown are representative of three independent experiments with consistent results.

**Figure 3 jof-09-00499-f003:**
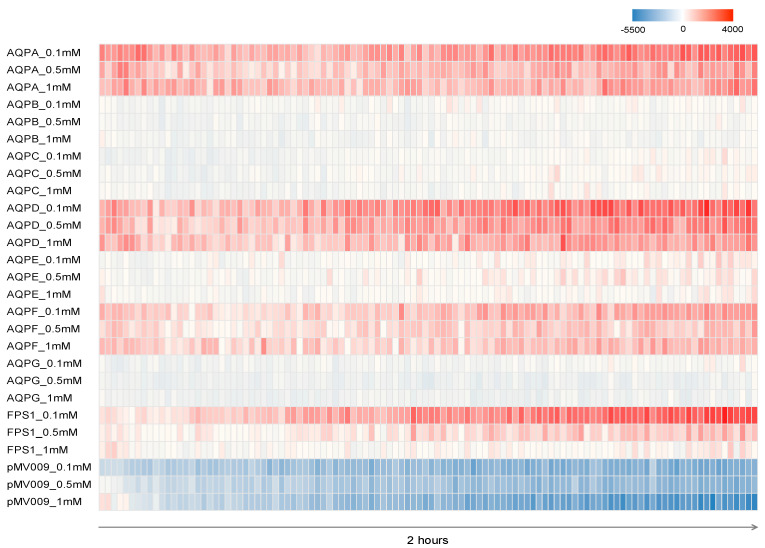
Hydrogen peroxide-dependent relative fluorescence in cultures of yeast expressing *A. niger* AQPs. Measurements of AQP-mediated hydrogen peroxide transport across membranes using the reactive fluorescent dye DCFHDA. Fluorescent intensity was measured for 2 h.

**Figure 4 jof-09-00499-f004:**
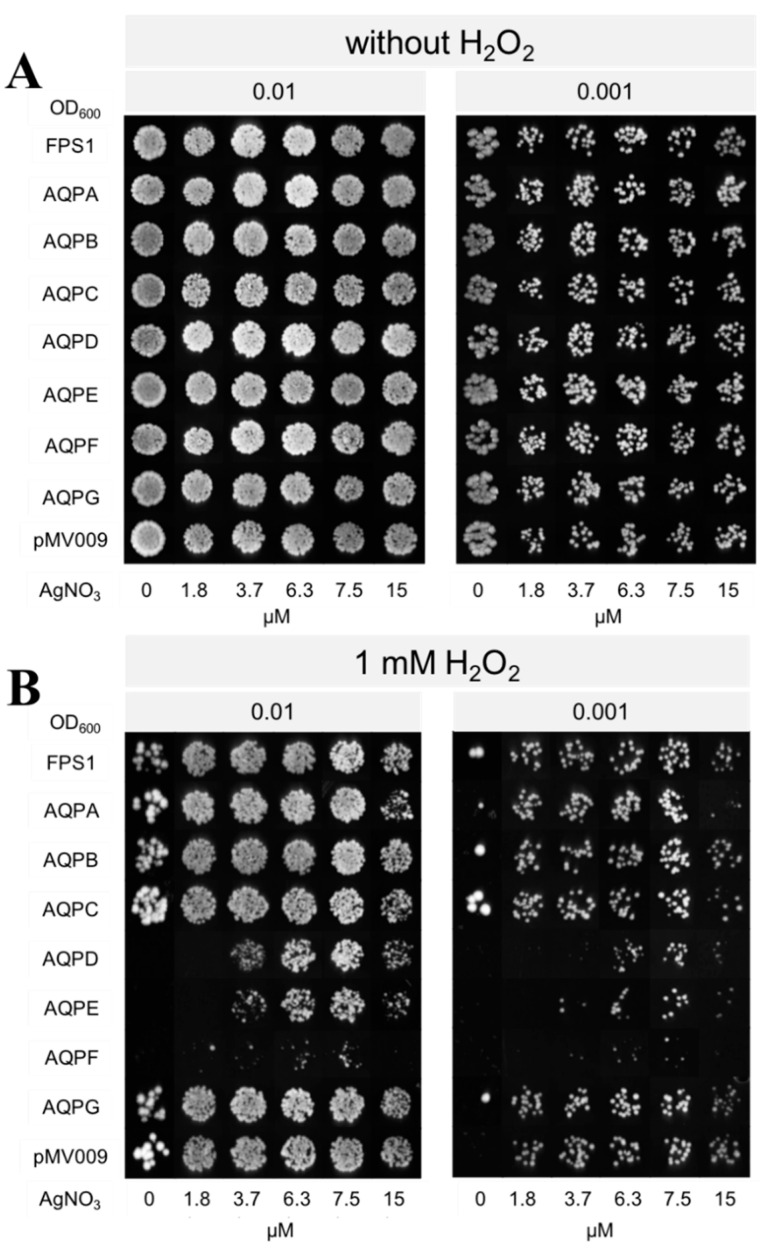
Phenotypic growth assay of *S. cerevisiae* strains expressing *A. niger* AQPs on AgNO_3_-containing agar media in the presence and absence of hydrogen peroxide. Overnight yeast cultures were diluted to an OD_600_ of 0.1, and subsequently, serial dilutions were done from this one. The 5 µL of cell suspensions were spotted on agar plates containing various AgNO_3_ concentrations (0–15 µM). Growth phenotype was recorded after incubating for 6 days at 30 °C. The data shown are representative of three independent experiments with consistent results. (**A**) No hydrogen peroxide added. (**B**) In the presence of 1 mM hydrogen peroxide.

**Figure 5 jof-09-00499-f005:**
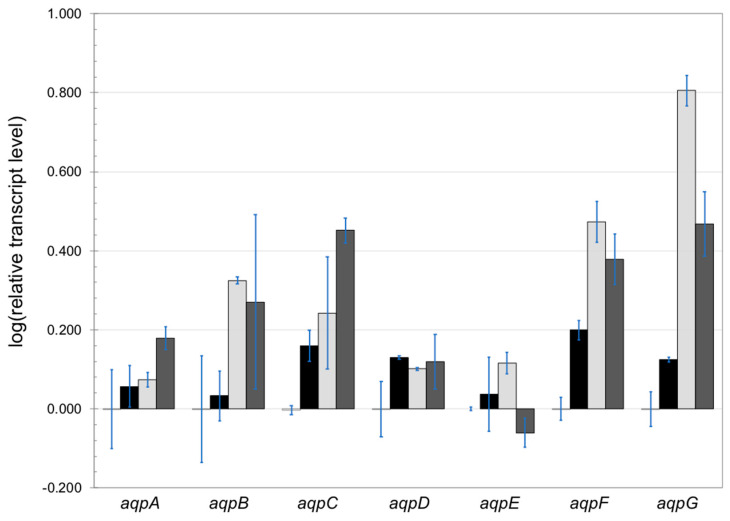
Expression of *A. niger aqp* genes under varying concentrations of hydrogen peroxide. Samples were taken 3 h after mycelium transfer to a medium containing various concentrations of hydrogen peroxide: from left to right: no hydrogen peroxide added, 2 mM (black), 10 mM (light gray), and 75 mM (dark gray) hydrogen peroxide. The expression analyses were performed by RT-qPCR. The transcript level of the histone-like gene was used to normalize transcript levels. Results are represented in relative transcript ratio in logarithmic scale (log10) with means of two biological replicates.

**Figure 6 jof-09-00499-f006:**
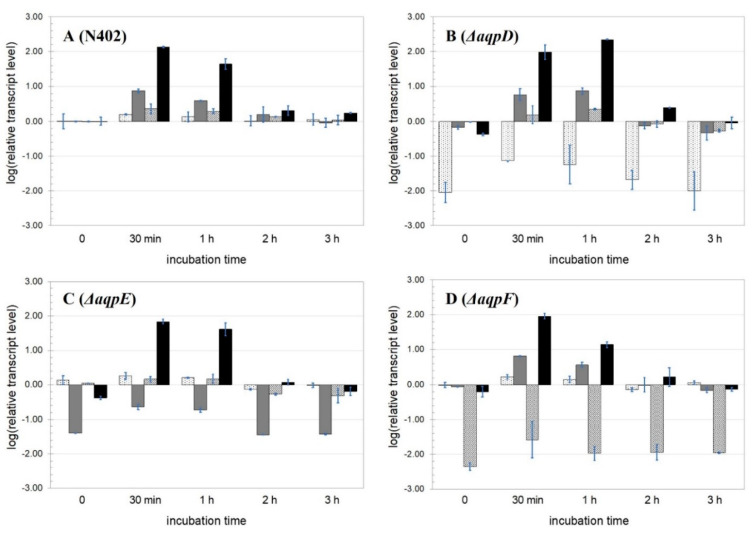
**Expression analysis of *A. niger* wild-type and AQP knock-out strains under hydrogen peroxide-inducing conditions.** Results of strain N402 (**A**), Δ*aqpD* (**B**), Δ*aqpE* (**C**), and Δ*aqpF* (**D**). Samples were taken before (0) or at various times after the addition of 10 mM hydrogen peroxide. The expression analyses were performed by RT–qPCR with primers specific for *aqpD* (white background filled with dots), *aqpE* (gray bar), *aqpF* (white background filled with stripes), and *goxC* (black). The *goxC* gene was used as a proxy for hydrogen peroxide uptake. The data presented is the mean of two biological replicates using a logarithmic scale (log10).

**Figure 7 jof-09-00499-f007:**
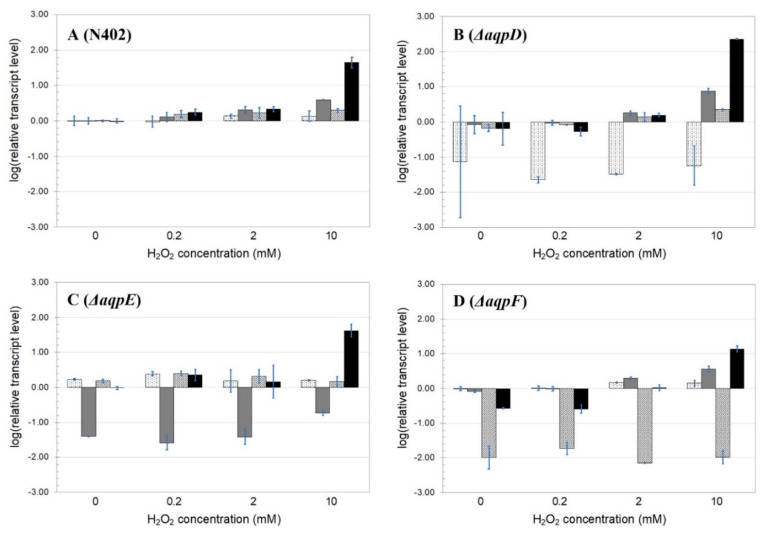
**Expression analysis of *A. niger* AQPs knock-out strains under varying concentrations of hydrogen peroxide**. Results of the strain N402 (**A**), Δ*aqpD* (**B**), Δ*aqpE* (**C**), and Δ*aqpF* (**D**). Samples were taken one hour after the addition of 0–10 mM hydrogen peroxide. The expression analyses were performed by RT-qPCR with primers specific for *aqpD* (white background filled with dots), *aqpE* (gray bar), *aqpF* (white background filled with stripes), and *goxC* (black). *goxC* gene expression was used as a proxy for hydrogen peroxide uptake. The data presented is the mean of two biological replicates using a logarithmic scale (log10).

**Table 1 jof-09-00499-t001:** Strains used in this study.

Strain	Remarks	Reference
*A. niger* ATCC 64974 (N402)		[27]
*A. niger* MA164.9	*kusA*::DR-*amdS*-DR, *pyrG*^−^	[28]
*A. niger* MA164.9 Δ*aqpD*		This study
*A. niger* MA164.9 Δ*aqpE*		This study
*A. niger* MA164.9 Δ*aqpF*		This study
*S. cerevisiae* BY4741	*MATa his3*Δ*1 leu2*Δ*0 met15*Δ*0 ura3*Δ*0*	[30]
*S. cerevisiae* BY4741 *aqpA*		This study
*S. cerevisiae* BY4741 *aqpB*		This study
*S. cerevisiae* BY4741 *aqpC*		This study
*S. cerevisiae* BY4741 *aqpD*		This study
*S. cerevisiae* BY4741 *aqpE*		This study
*S. cerevisiae* BY4741 *aqpF*		This study
*S. cerevisiae* BY4741 *aqpG*		This study
*S. cerevisiae* BY4741 *FPS1*		This study
*S. cerevisiae* BY4741 pMV009	Control plasmid expressing GFP instead of an *aqp* gene	This study

**Table 2 jof-09-00499-t002:** List of *A. niger* N402 aquaporin genes.

Name	Gene Locus Tag	Protein ID	Predicted Class
*aqpA*	ATCC64974_2160	SPB42568.1	orthodox AQP
*aqpB*	ATCC64974_25380	SPB44895.1	XIP
*aqpC*	ATCC64974_26080	SPB44966.1	Yfl054-like
*aqpD*	ATCC64974_52260	SPB47598.1	γ^2^-AQGP
*aqpE*	ATCC64974_71680	SPB49549.1	Yfl054-like
*aqpF*	ATCC64974_75350	SPB49917.1	XIP
*aqpG*	ATCC64974_111300	SPB53530.1	Unclassified

## Data Availability

Data are contained within the article or Appendix A.

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
