# Peer review of "Identification of Aspergillus niger Aquaporins Involved in Hydrogen Peroxide Signaling"

_jof, 2023, doi:10.3390/jof9040499_

Round 1

Reviewer 1 Report

In this paper, AQPs were identified in Aspergillus niger, and functionally characterized by heterologous expression in yeast and gene transcriptional expression testing in ACP gene knockouts. In general, plenty of work has been done and the manuscript is well-written. A few minors revisions needed be done as following :

1.     In the introduction part, the significance of studying Aquaporin proteins mediated hydrogen peroxide uptake is not clear.

2.  Line 88:  “ pWay-pyrA plasmid” or  “pWay-pyrG plasmid” ?

3. The authors mentioned that “Upon addition of hydrogen peroxide to the medium transcript levels, all AQP  genes increased (Figure 5)”, however, in the Figure5, the expression level of AQPE is decreased under 75 mM condition. In addition , there is a grammar in this sentence it should be revised as : “Upon addition of hydrogen peroxide to the medium,  transcript levels of all AQP  genes increased (Figure 5)”.

4. It would be better to investigate other hydrogen peroxide responsive genes in the AQP gene mutants.

Author Response

Thanaporn Laothanachareon                            

National Center for Genetic Engineering and Biotechnology, Thailand

Peter J. Schaap

Wagenigen University & Research, The Netherlands

To

Journal of Fungi

Dear Editors and reviewers,

On behalf of all authors, thank you for comments and suggestions to our manuscript, entitled “Identification of Aspergillus niger aquaporins involved in hydrogen peroxide signaling”, to Journal of Fungi”. We responded to these comments and suggestions as you can see in the attached files.

Yours sincerely,

Thanaporn Laothanachareon

National Center for Genetic Engineering and Biotechnology, Thailand.

and

Peter J. Schaap

Wagenigen University & Research, The Netherlands

Reply to the review report

In this paper, AQPs were identified in Aspergillus niger, and functionally characterized by heterologous expression in yeast and gene transcriptional expression testing in ACP gene knockouts. In general, plenty of work has been done and the manuscript is well-written. A few minors revisions needed be done as following :

  1. In the introduction part, the significance of studying Aquaporin proteins mediated hydrogen peroxide uptake is not clear.

As indicated in the Introduction section “gluconic acid is the product of an extracellular conversion of glucose in a collaboration of three extracellular enzymes: glucose oxidase (GOx), lactonase, and catalase (Witteveen et al., 1993). At an extracellular pH between 4.5 and 6.5 and in the presence of sufficient amounts of oxygen, glucose oxidase catalyses the oxidation of glucose to gluconolactone and hydrogen peroxide.” 

Glucose oxidase is an extracellular enzyme and recently, we studied the role of hydrogen peroxide byproduct in the regulation of the A. niger (GOx) expression system (Laothanachareon et al., 2018) and showed that inducer of the GOX-system is the hydrogen peroxide byproduct of glucose oxidase which suggests facilitated diffusion of hydrogen peroxide across the plasma membrane mediated by aquaporins. We have now made this more explicit and changed line 63 accordingly

Line 63 In A. niger, the hydrogen peroxide by-product acts as a second messenger triggering, in a feed-forward loop, the expression of the GOx system (Laothanachareon et al., 2018) which suggests facilitated diffusion of hydrogen peroxide across the plasma membrane mediated by aquaporins.

  1. Line 88:  “ pWay-pyrA plasmid” or  “pWay-pyrG plasmid” ?

The pyrG gene was amplified from A. oryzae and used as a selectable marker to replace the aqp genes, The pWay-pyrA plasmid is a complete plasmid, constructed by amplifying pyrA gene from A. niger and ligated to the vector. This plasmid was used as a control for transformation. The approach taken is similar to Laothanachareon et al., 2018 (doi: 10.3389/fmicb.2018.02269.) and Laothanachareon et al, 2021 (https://doi.org/10.3390/jof7060409).

  1. The authors mentioned that “Upon addition of hydrogen peroxide to the medium transcript levels, all AQP genes increased (Figure 5)”, however, in the Figure 5, the expression level of AQPE is decreased under 75 mM condition. In addition , there is a grammar in this sentence it should be revised as : “Upon addition of hydrogen peroxide to the medium,  transcript levels of all AQP  genes increased (Figure 5)”.

This sentence was improved as the suggestion. (L328)

  1. It would be better to investigate other hydrogen peroxide responsive genes in the AQP gene mutants.

Previously we have shown there is a direct link of the transcription levels of the goxC gene encoding glucose oxidase and the uptake of hydrogen peroxide (Laothanachareon et al., 2018 (doi: 10.3389/fmicb.2018.02269.) Our previous study also showed that catR (encoding catalase R), sodA, (encoding superoxide dismutase A) and gstA (encoding glutathione-S-transferase A) genes respond to various hydrogen peroxide concentrations. However, their transcription levels were lower than that of goxC gene.

Reviewer 2 Report

The article is very interesting, highlighting the role of channel protein -aquaporins. So, before it could be considered for publication, I have some quires which authors need to incorporate and extensively revise their manuscript.

The abstract section needs to give proper information. Abstract means a full-fledged summary highlighting the information and topics the manuscript covers. Please revise the abstract (it needs to rephrase and rewrite some sentences). Also, highlight essentialities and future perspectives of the study. Avoid introduction and references in the abstract. 

Note that the abstract should begin with a brief but precise statement of the problem or issue, followed by a description of the research method and design, major findings, and conclusions.

Add more background information so the readers can understand and summarize the main points.

Section Introduction

The introduction looks fine. However, the authors need to summarize well the main points in the last paragraph. 

Section results and discussion 

Detailed; however, it needs more supporting references. Many statements need references. 

Section conclusion

The section needs to be included; the authors should write it elaborately and highlight the study's importance and future directions with possible limitations.

Overall, the language needs more polishing. Besides, improve the quality of graphs. 

Author Response

Thanaporn Laothanachareon                            

National Center for Genetic Engineering and Biotechnology, Thailand

Peter J. Schaap

Wagenigen University & Research, The Netherlands

To

Journal of Fungi

Dear Editors and reviewers,

On behalf of all authors, thank you for comments and suggestions to our manuscript, entitled “Identification of Aspergillus niger aquaporins involved in hydrogen peroxide signaling”, to Journal of Fungi”. We responded to these comments and suggestions as you can see in the attached files.

Yours sincerely,

Thanaporn Laothanachareon

National Center for Genetic Engineering and Biotechnology, Thailand.

and

Peter J. Schaap

Wagenigen University & Research, The Netherlands

Reply to the review report

The article is very interesting, highlighting the role of channel protein -aquaporins. So, before it could be considered for publication, I have some quires which authors need to incorporate and extensively revise their manuscript.

The abstract section needs to give proper information. Abstract means a full-fledged summary highlighting the information and topics the manuscript covers. Please revise the abstract (it needs to rephrase and rewrite some sentences). Also, highlight essentialities and future perspectives of the study. Avoid introduction and references in the abstract. 

Note that the abstract should begin with a brief but precise statement of the problem or issue, followed by a description of the research method and design, major findings, and conclusions.

Add more background information so the readers can understand and summarize the main points.

We thank the reviewer for the suggestions and have adapted the Abstract section accordingly.

Section Introduction

The introduction looks fine. However, the authors need to summarize well the main points in the last paragraph. 

We thank the reviewer for the suggestions.

The last paragraph now reads

By screening the genome sequence of A. niger strain ATCC 64974 (N402) seven putative AQPs were identified.  Experiments were designed to observe aquaporin facilitated hydrogen peroxide transport. Yeast strains expressing individual AQPs were evaluated by monitoring growth under various hydrogen peroxide concentrations and by fluorescence assays. Apart from the yeast phenotypic growth assays, A. niger AQP knock-out strains were constructed, and AQP transcript levels were monitored in wild-type and AQP knock-out strains before and after the addition of hydrogen peroxide to the medium. From the yeast experiments, three AQPs were identified with a potential role in hydrogen peroxide membrane transport.  Integration of these results with A. niger expression data suggests that AQPF may play an important role in facilitating hydrogen peroxide transport across the A. niger cellular membrane.

Section results and discussion 

Detailed; however, it needs more supporting references. Many statements need references. 

The manuscript contains 53 supporting references. We checked the manuscript for any omissions. We think it is fine now. 

Section conclusion

The section needs to be included; the authors should write it elaborately and highlight the study's importance and future directions with possible limitations.

Overall, the language needs more polishing. Besides, improve the quality of graphs. 

We did not add a conclusion section but instead summarized the results at the end of the Discussion section.

Reviewer 3 Report

The manuscript (Identification of Aspergillus niger aquaporins involved in hydrogen peroxide signaling) is interesting and completes their previous work to explain the role of hydrogen peroxide in regulating the expression of glucose oxidase.

Please avoid citing references in the abstract.

Please update the old references with recent ones.

Please provide the accession numbers of the isolates

Figure 5 should be expressed in fold change.

L459 Please avoid repeating the identification of hydrogen peroxide, as you already explained in the abstract and introduction.

Please follow the Journal guidelines, especially when you cite references in the text.

Minor comments

L23 I think you mean Seven proteins. Add the subject.

L33 Change to their specific roles in fungi are, however, less understood

L36 Change to facilitating the transport of the hydrogen peroxide signal.

L42 and 43 Rewrite in more understandable way. I suggest the following (Aspergillus niger is a potent cell factory that is successfully employed to create organic acids including citric-, oxalic-, and gluconic acids).

L89 Change to strains before and after the addition

L120 Change to applied to the phylogenetic analysis

L138 Change to templates for the amplification

L159 Change to and subsequently, serial dilutions

L161 Change to suspensions was spotted on the solid

L190 Please keep uniformity in the text and change to the knock-out strains

L223 Change to AQPG, presenting

L271 Change to a slight decrease in the fluorescence signal.

L286 Figure 2 legend change to (various hydrogen peroxide concentrations)

L294 Grammatical check, I suggest changing to sequences that are targets

L298 Change to the previously reported

L300 Change to (The hydrogen peroxide-induced growth phenotype of the yeast strains expressing aqpD and aqpE was considerably improved).

L416 Change to in the liquid medium

L447 Change to plants, and fungi

L451 Change to facilitate the transfer

L453 Change to boric acid, and ammonia

L465 Change to upon the addition

Author Response

Thanaporn Laothanachareon                            

National Center for Genetic Engineering and Biotechnology, Thailand

Peter J. Schaap

Wagenigen University & Research, The Netherlands

To

Journal of Fungi

Dear Editors and reviewers,

On behalf of all authors, thank you for comments and suggestions to our manuscript, entitled “Identification of Aspergillus niger aquaporins involved in hydrogen peroxide signaling”, to Journal of Fungi”. We responded to these comments and suggestions as you can see in the attached files.

Yours sincerely,

Thanaporn Laothanachareon

National Center for Genetic Engineering and Biotechnology, Thailand.

and

Peter J. Schaap

Wagenigen University & Research, The Netherlands

Reply to the review report

The manuscript (Identification of Aspergillus niger aquaporins involved in hydrogen peroxide signaling) is interesting and completes their previous work to explain the role of hydrogen peroxide in regulating the expression of glucose oxidase.

Please avoid citing references in the abstract.

            We have revised the abstract accordingly.

Please update the old references with recent ones.

Please provide the accession numbers of the isolates

The accession number was provided to A. niger N402.

Figure 5 should be expressed in fold change.

The figure 5, 6, and 7 were calculated and interpreted following Sloothaak et al, 2016 (doi: 10.1186/s13068-016-0564-4.)

L459 Please avoid repeating the identification of hydrogen peroxide, as you already explained in the abstract and introduction.

The redundant sentence was removed. (L468-470)

Please follow the Journal guidelines, especially when you cite references in the text.

We have implemented the journal guidelines with respect to the references.

Minor comments

L23 I think you mean Seven proteins. Add the subject.

The subject was added to “Seven AQPs were found…”

L33 Change to their specific roles in fungi are, however, less understood

Commas were added as suggested.

L36 Change to facilitating the transport of the hydrogen peroxide signal.

These sentences were truncated.

L42 and 43 Rewrite in more understandable way. I suggest the following (Aspergillus niger is a potent cell factory that is successfully employed to create organic acids including citric-, oxalic-, and gluconic acids).

This sentence was rewritten as the suggestion. (L46-47)

L89 Change to strains before and after the addition

This sentence was improved as the suggestion. (L96)

L120 Change to applied to the phylogenetic analysis

This sentence was improved as the suggestion. (L142)

L138 Change to templates for the amplification

This sentence was improved as the suggestion. (L159)

L159 Change to and subsequently, serial dilutions

This sentence was improved as the suggestion. (L180)

L161 Change to suspensions was spotted on the solid

L190 Please keep uniformity in the text and change to the knock-out strains

This sentence was improved as the suggestion. (L205)

L223 Change to AQPG, presenting

This sentence was improved as the suggestion. (L245)

L271 Change to a slight decrease in the fluorescence signal.

This sentence was improved. (L285)

L286 Figure 2 legend change to (various hydrogen peroxide concentrations)

This sentence was improved as the suggestion. (L307)

L294 Grammatical check, I suggest changing to sequences that are targets

This sentence was improved as the suggestion. (L316)

L298 Change to the previously reported

This sentence was improved as the suggestion. (L319)

L300 Change to (The hydrogen peroxide-induced growth phenotype of the yeast strains expressing aqpD and aqpE was considerably improved).

This sentence was improved as the suggestion. (L320-321)

L416 Change to in the liquid medium

This sentence was improved as the suggestion. (L438)

L447 Change to plants, and fungi

This sentence was improved as the suggestion. (L470)

L451 Change to facilitate the transfer

This sentence was improved as the suggestion. (L475)

L453 Change to boric acid, and ammonia

This sentence was improved as the suggestion. (L476)

L465 Change to upon the addition

This sentence was improved as the suggestion. (L489)

Round 2

Reviewer 2 Report

The authors have addressed most of the comments wherever necessary.

It is important to ensure that the comments and feedback are carefully considered and addressed to improve the quality and credibility of the research. The end of discussion has no end.